# Development of a new methodology for the determination of PET microplastics in sediment, based on microwave-assisted acid digestion

**Marco Perez**[1], **Sonnia Parra**[1], **Cristofher Ferrada**[1], **Manuel Bravo**[1], **Pablo A. Perez**[2], **Waldo Quiroz**[1] *

1 Instituto de Química, Pontificia Universidad Católica de Valparaíso, Valparaíso, Chile, 2 Departamento de Ciencias Farmacéuticas, Facultad de Ciencias, Universidad Católica Del Norte, Antofagasta, Chile

* waldo.quiroz@pucv.cl

**Data Availability Statement:** All relevant data are within the manuscript and its Supporting Information file which can be acces in thus URL

## Abstract

Analytical methods for the determination of microplastics in sediments typically involve matrix drying, sieving, grinding, and flotation as part of the sample treatment. However, the real need for these steps and analytical validation studies are scarce. This work proposes a method that avoids the drying, sieving, and flotation procedures by using a direct acid attack of $HNO_3/HCl$ (3:1) on wet sediment samples, assisted by microwave digestion. For detection, induced fluorescence using a UV camera, with Nile Red (NR) as the fluorophore and a cell phone camera for image capture were used. The results showed that when the digestion temperature was raised to 120°C, PET recovery decreased due to plastic particle fusion. However, at 60°C, microwave digestion resulted in a 97% recovery of PET particles, eliminating chitin interference and canceling cellulose fluorescence without the need for flotation. This method proved effective for monitoring plastic microparticles in sediments from the Loa River, Chile, revealing that the river is predominantly contaminated with PET microparticles, particularly upstream in the Taira area.

## Introduction

Marine sediments stand out as one of the most studied environmental samples in relation to microplastic pollution. According to sciencedirect.com, approximately 4956 references on microplastics (MPs) in marine sediments between 2020 and 2023 (2020: 849; 2021: 1,174; 2022: 1,601; 2023: 1,332). Clearly, numerous efforts are reflected in scientific publications, providing insights to understand the impact of microplastic pollution on the marine ecosystem.

In efforts to better understand this type of contamination, several methodologies are reported for both sample pretreatment and extraction of MPs from sediments. The most common methods for sediment pretreatment are drying and sieving; In the extraction stage of the MPs, we find extraction by density (flotation) and digestion. Various studies employ these methods in different configurations, depending on the specific requirements of the analysis [1–5].

https://osf.io/3rmqu/?view_only=f786d2c7739a40a2801ad39dbe72e407.

**Funding:** This work has the financial support of the Chilean government through its agency "Agencia Nacional de Investigación y Desarrollo (ANID)" via the "Fondo Nacional de Desarrollo Científico y Tecnológico (Fondecyt)" project 1230585. The funders had no role in study design, data collection and analysis, decision to publish, or preparation of the manuscript".

**Competing interests:** The authors have declared that no competing interests exist.

However, it has been reported that many of these pretreatment methods lack proper validation [6] which may introduce significant errors, particularly for polymeric microparticles with high densities. This is reflected in the low recovery percentages as is the case with PET [7]. For instance, a review by Lu et al [1] analyzed 183 studies on MPs in freshwater and sediment ecosystems, and highlighted a lack of standardized protocols for quality assurance.

The absence of standardized analysis protocols has led to significant methodological variability across studies and poor validation practices. Regarding the pretreatment step, Phuong et al. [3] in 2021, conducted a critical review of 70 studies on the determination of PMs in marine sediments finding that only 50% of these studies performed the analysis with dry sediments. The review identified three drying techniques were identified: first, freeze-drying, reported in three studies; secondly, the drying of sediments at temperatures between 50 and 80°C for periods ranging from 16 to 72 hours, reported in 26 studies; and third, air and sun drying, mentioned in 6 studies. Similarly, Prata et al. [6] note that only 9 out of 20 studies report MPs concentrations based on dry weight.

Regarding density separation of MPs, Phuong et al. [3] reported that 93% of the studies used flotation with solutions of varying densities. The most commonly used solutions were sodium chloride (NaCl), zinc chloride ($ZnCl_2$), and sodium iodide (NaI) with densities typically ranging from 1.18to 1.20, 1.37 to 1.80, and 1.43to1.80 g cm$^{-3}$ respectively. It should be noted that of the 65 studies that used the flotation stage, only 20 performed recovery studies. Of those, 8 reported recoveries greater than 90% for low density MPs such as PP and PE. However, low recoveries (<90%) were observed for denser polymers like PVC and PET. In a study by Vermeiren et al. [7] recoveries for PVC and PET ranged from 82% to 86%.

Chemical digestion is commonly used for the selective determination of MPs through the mineralization of the sample matrix [7–12]. However, MPs can be destroyed in this process, especially in extreme conditions. Therefore, it is necessary to find a balance between the degradation of sample matrix and the integrity of MPs. Phuong et al. [3] report the use of acidic, basic, and enzymatic digestions. Enzymatic digestion best preserves the integrity of MPs, but its cumbersome nature makes it less attractive for routine testing. Studies carried out in the sediment matrix are scarce and most cases apply digestion with $H_2O_2$, NaOH or the use of the Fenton´s reagent [3–6]. Duan et al. [9] reported using concentrated $HNO_3$ to digest sediments rich in organic matter, which proved highly effective for organic matter removal but caused color changes in PE and PP MPs, reducing the selectivity for PE and PET detection.

While some studies have attempted to validate their methodologies [7, 13, 14], most methods found in the literature lack adequate validations for each methodological step [6]. This lack of validation undermines the reliability of methodologies. Furthermore, it remains unclear whether the combination of these pretreatment and extraction techniques is always necessary, as many steps may be redundant if the detection method is sufficiently sensitive and selective.

Polyethylene terephthalate (PET) is primarily produced for the manufacturing of plastic bottles and containers [15]. PET and polystyrene (PS) together account for 79.1% of the polymers detected at various stages of wastewater treatment plants [16]. Additionally, a significant presence of floating plastic bottles has been reported in the Loa River, Chile [17]. Therefore, the aim of this study is to develop a fast, simple, and cost-effective analytical methodology for the reliable determination of PET MPs in Loa River sediments. This method utilizes microwave-assisted acid digestion ($HNO_3$/HCl, 3:1 v/v) applied directly to wet sediment samples.

## Materials and methods

### Reagents

The reagents and solvents used were Scharlau HPLC grade acetone, Supelco suprapure hydrochloric acid, Supelco suprapure nitric acid (65%), Milli-Q laboratory water, and Nile Red Sigma Aldrich.

Analytical-grade chitin and purified marine sand (both from Sigma Aldrich) and crude cellulose were used to simulate the sedimentary matrix.

The MPs used in the experiments represent the polymers most found in the marine environment: polyethylene terephthalate (PET), low-density polyethylene (LDPE), polystyrene (PS) and polypropylene (PP)[3]. MPs were made from virgin plastic resins of each type of polymer. A grinding system was implemented to ensure that the resin is free from contact with any other polymer. The grinding process involves chopping at 5-second intervals with 10-second breaks to prevent heating and possible structural changes in each polymer. Subsequently, the material is screened through 1.0, 0.5 and 0.15 mm mesh, resulting in two working fractions (1–0.5 mm and <0.5–0.15 mm), which are stored in stainless steel containers.

### Sample collection

River sediment samples were collected from the Loa River Basin, located in northern Chile, within the Antofagasta region. Spanning 440 kilometers, the Loa is Chile's longest river and the primary source of freshwater in the world's driest desert. Specifically, samples were taken from three locations along the basin:

- Taira (21˚55'32.48"S, 68˚36'26.06"W), representing the upper Loa.

- Chiu-Chiu (22˚22'07.5"S, 68˚39'12.6"W), representing the middle Loa.

- The river's mouth (21˚25'29.0"S, 70˚02'33.9"W), representing the lower Loa.

River sediment samples were collected from a depth of 5–10 cm. Sampling sites are shown in Fig 1.

Based on Chilean law, all sea and river shores are publicly accessible, so no permits of any kind are required to access them. Furthermore, the Loa River in this study is not under any special environmental protection.

### Sample treatment

The Fig 2 flowchart of the method for determining PET MPs in sediment samples. Those steps commonly described in the literature for the analysis of MPs in sediments, such as drying, sieving, and extraction of MPs due to density differences, were eliminated.

This methodology mainly consists of applying microwave-assisted acid digestion directly to the wet sediment. In this stage, the optimal conditions of digestion temperature for matrix removal, stability of PET MPs, selectivity as a function of mass recovery and particle count were evaluated.

The digestion step was carried out with 20 mL of a mixture of $HNO_3$/HCl in a ratio of 3:1 v/v, in a closed microwave-assisted system (ETHOS EASY milestone connect with a capacity of 15 digestion vials) for one hour. This combination of acids was chosen because they are commonly used in sediment digestion for metal quantification and are frequently used reagents in chemical analysis laboratories [18, 19]. Three digestion temperatures (60, 80 and 120˚C) were studied with different synthetic components of the sedimentary matrix.

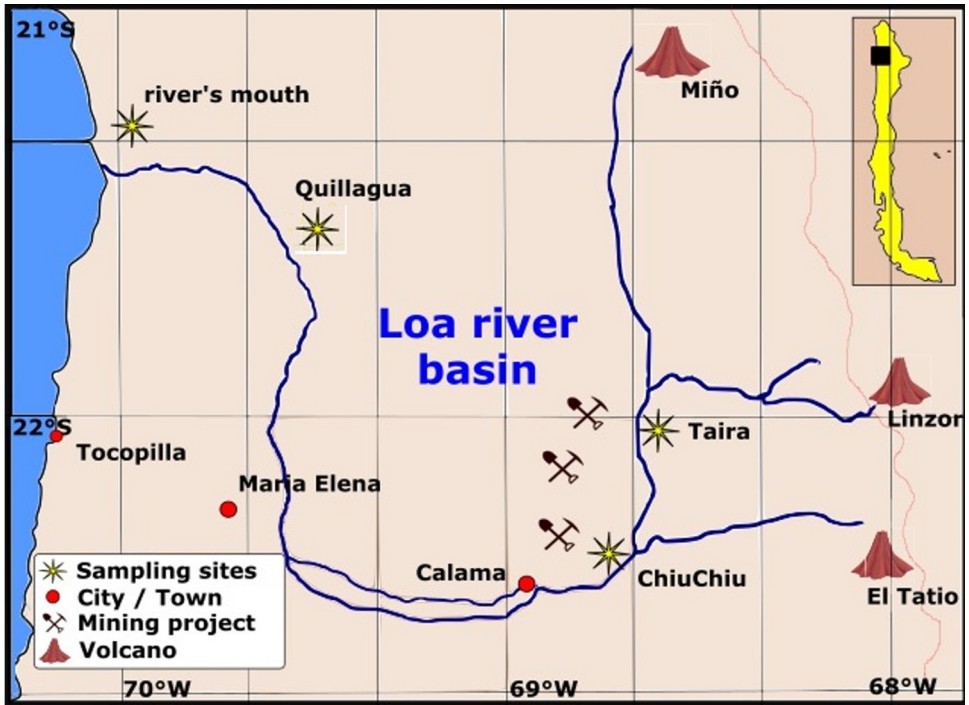

**Fig 1. Sampling sites for sediment collection along the Loa River, Chile.** Figure adapted from OpenStreetMap.

Preliminary experiments were carried out separately for chitin, cellulose, and polymer micro-particles from PET, LDPE, PS and PP.

After digestion, the samples are filtered by a 0.45 μm nitrocellulose membrane and placed in a Petri dish, 100 mm in diameter, to be dried in an incubator, for about 1 hour at 60°C, for their subsequent detection stage.

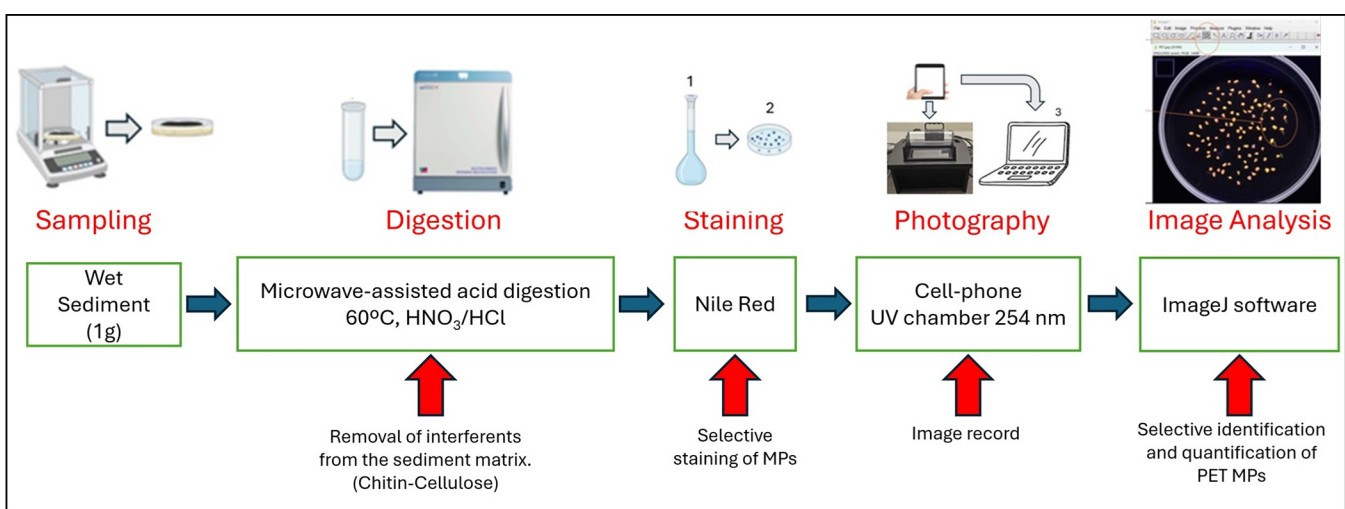

**Fig 2. Flowchart of the method for determining PET MPs in sediment samples.** The red arrows indicate the analytical purpose of each stage.

## Detection step

The staining step was implemented as described by Lv et al. [20]. A solution of Nile red (NR) was prepared at a concentration of 10 mg L$^{-1}$, at 1% v/v in acetone of 99.5% v/v. The solid components of the filtered digest were incubated in petri dishes, 100 mm in diameter, with 20 mL of NR at 50˚C for a period of 10 minutes, covered with aluminium foil. After the incubation period, the solution was rapidly cooled in an ice water bath at 0˚C for 20 minutes. The application of this staining method, using a cycle of high and low temperatures, is based on the physical properties of thermal expansion and contraction in plastics. At high temperatures, dye molecules can penetrate the microplastics and then when the temperature is rapidly reduced, the structure contracts, encapsulating the dye molecules. Empirical observations using this method have shown us that the particles exhibit more intense and longer-lasting fluorescence.

The solution was then filtered with a 0.45 μm nitrocellulose membrane. The filter was stored on the same glass plate, which was covered with aluminium foil and placed in a drying oven at 60˚C for two hours. After the drying time, the plates were taken to the fluorescence booth (Spectroline model CM-10ª) equipped with UV lamp and excitation filter at 254 nm [6]. The fluorescence images were captured with a Samsung Galaxy S20 FE phone, model SM-G780G. Image processing for particle counting was performed manually with ImageJ software.

## Trueness determination

Mass recoveries for each polymer and microparticle count were evaluated by recovery tests under acidity and temperature conditions. In addition, the elimination capacity of chitin and cellulose was studied. This validation consisted of the evaluation of the recovery of both the particle count and the mass recovery of MPs. Approximately 100 particles of each polymer were subjected to the methodology implemented for the recovery of the number of particles. These experiments were performed in triplicate to verify the stability MPs under the defined conditions of digestion, mixing particles of each polymer to evaluate the selectivity. Given the heterogeneity in the distribution of microplastics in the environment, recovery studies on real samples cannot be conducted using naturally contaminated samples, as the native microplastic content varies between sample aliquots and cannot be subtracted mathematically. For this reason, in order to conduct a recovery study on a real sedimentary matrix, the sample was thermally treated to calcine the native microplastics. by a muffle furnace calcination procedure at 800˚C, creating a sedimentary matrix as close to real conditions as possible, but without native microplastic contamination that would interfere with recovery studies. Mass recovery experiments were performed in triplicate and at four mass levels: 0.025 g, 0.050 g, 0.075 g and 0.100 g to assess the effects of digestion on the potential degradation of MPs.

## Statistical analysis

The data systematization and analysis were conducted using Microsoft Excel. For statistical validation, Student's t-tests with a 95% confidence level were employed. To evaluate differences in concentration values between sampling sites, a t-test for datasets assuming unequal variances was applied, using the data analysis tools available within the same software.

The protocol described in this peer-reviewed article is published on protocols.io, https://dx.doi.org/10.17504/protocols.io.5jyl82b77l2w/v1 and is included for printing as supporting information file 1 with this article.

## Results and discussions

### Implementation of the staining step

Due to the heterogeneity of the sediment matrix, the staining study was conducted on the main organic interferences for fluorescence detection, specifically chitin and cellulose, as described by Maes et al. [14]. Additionally, the selectivity in the identification of PET MPs among the other three MPs was verified.

As a first step, staining was performed without the digestion step to fine-tune this process. Then the stained and digested components were compared to visualize changes in method selectivity.

Fig 3 shows the images of cellulose before (a) and after (b) the digestion process and with NR staining. It is important to note that, while chitin fluorescence is eliminated upon digestion, this is not entirely the case with cellulose. Although 30% of cellulose is removed, its fluorescence is turned off after digestion, as shown in Fig 3B.

On the other hand, Fig 4 shows NR-stained images of MPs corresponding to the four polymers used in this study at 254 nm and 365 nm excitation wavelength. According to the indications of Maes et al. [14], in Fig 3, based on the solvato-chromic properties of the NR and the different degrees of "polarity" of each of the four types of stained MPs, the fluorescence emission shifts to different wavelengths in the visible spectrum. This increases selectivity when identifying PET MPs in environmental matrices containing mixtures of these four polymers. By applying the digestion process, selectivity is further increased with respect to the analysis of PET MPs, as the interference caused by the PS MPs is eliminated as there is no fluorescent response to the excitation source used at 254nm. Finally, we can observe that the use of an excitation wavelength of 365 nm does not generate fluorescence that is sufficiently sensitive and selective for the identification of PET, LDPE, PP, and PS.

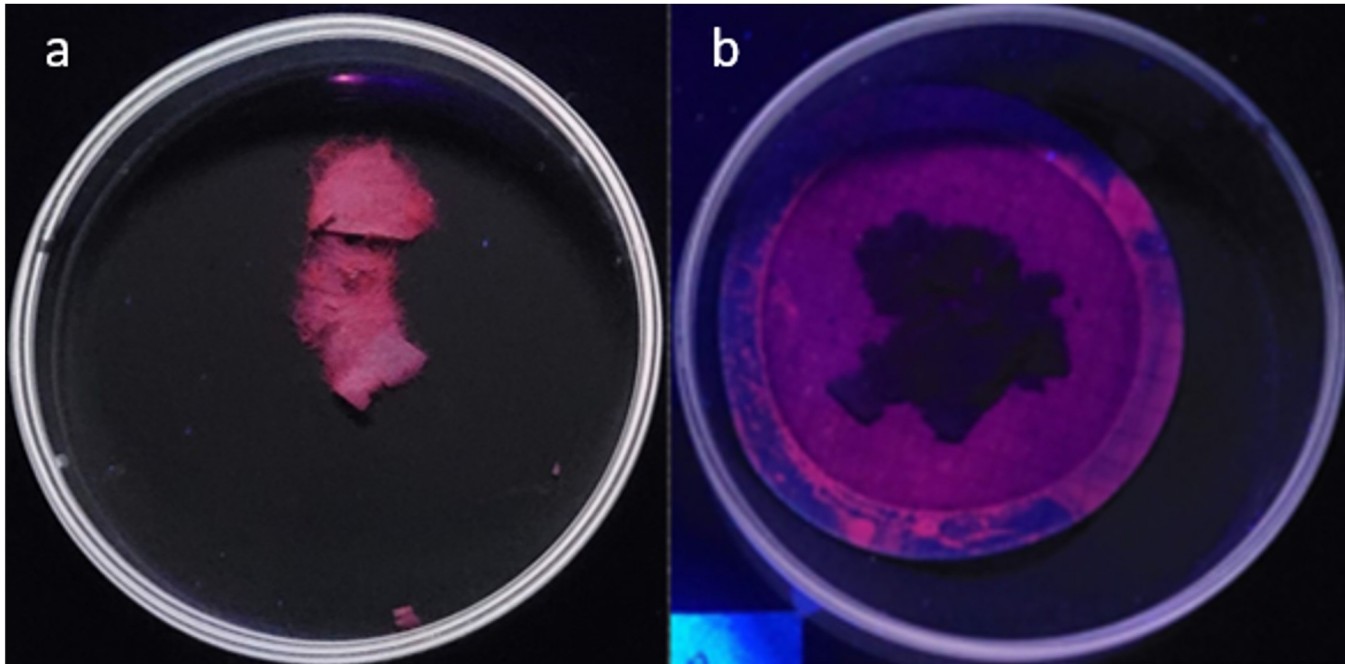

**Fig 3.** Results of staining of a) pre-digestion cellulose and b) post-digestion cellulose. Photographed at the excitation wavelength of 254 nm.

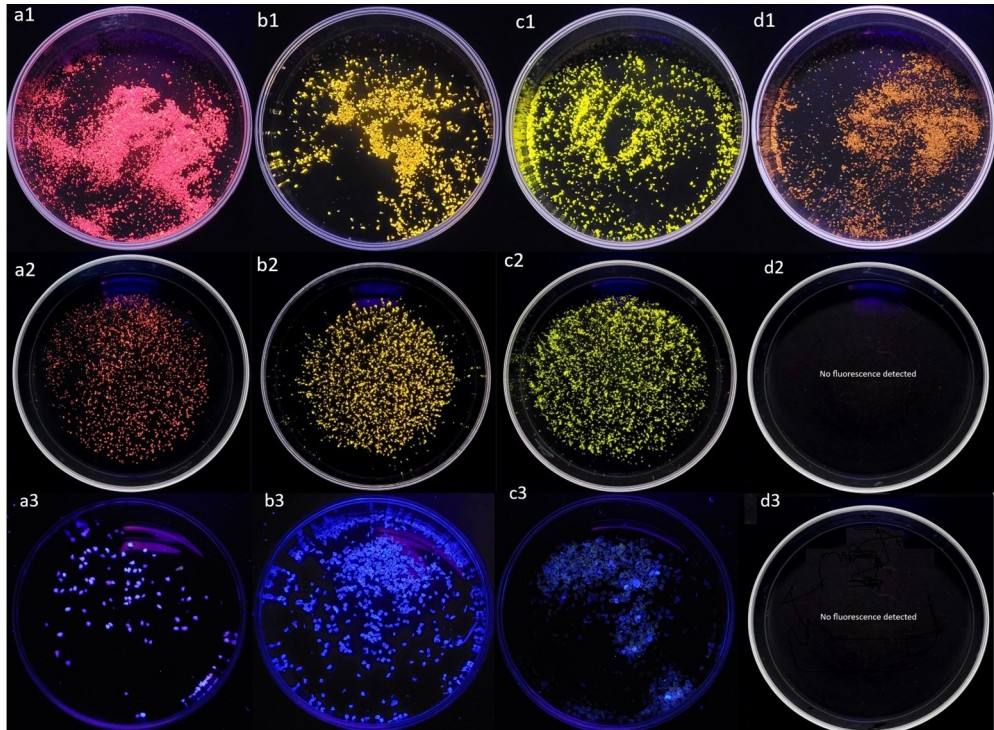

**Fig 4.** Results of the staining of the MPs used in this study, pre- digestion 254 nm a1) PET, b1) LDPE, c1) PP and d1) PS; post-digestion 254 nm a2) PET, b2) LDPE, c2) PP and d2) PS; post-digestion 365 nm a3) PET, b3) LDPE, c3) PP and d3) PS.

The results indicate that the staining step after digestion successfully eliminates PET interferences caused by cellulose and PS, as they do not exhibit a fluorescent response under the conditions of this methodology. Additionally, the method allows for differentiation between PET MPs and PP or LDPE MPs due to the distinct fuorescence colors of each polymer.

Given that the staining process involves multiple steps, including filtration and drying, it was necessary to assess the recovery rate of PET MPs.

### Effect of temperature digestion on mass recovery of MPs

Digestion experiments of the four types of MPs were performed at three different temperatures (60, 80 and 120°C) to determine the effect of temperature on MPs stability and recovery. The tests were performed in duplicate, using 0.5 g of each polymer and 0.5 g of each sediment component. The Fig 5 shows the percentages of mass recovery obtained for each polymer at the three temperatures studied. Raw data of Fig 5 can be found in S1 Table.

According to the results obtained in Fig 5, the best recovery percentages were obtained at 60 and 80°C. Digestion at 120°C presents mass loss, with values below 90% for PET and PS recoveries and a more significant variation in results for PP and LDPE. In addition, LDPE, and PP microparticles melt during digestion at 120°C, making it unfeasible to analyse MPs at this digestion temperature. Fig 6 shows the recovery of molten LDPE after digestion. This result is consistent with the study reported by Avio et al. [21], where the test of an acid digestion method verifies the degradation of the MPs of LDPE and PS.

The Fig 7 shows the results of digestion at temperature in the elimination of cellulose interference at three different digestion temperatures. The percentage of chitin elimination in

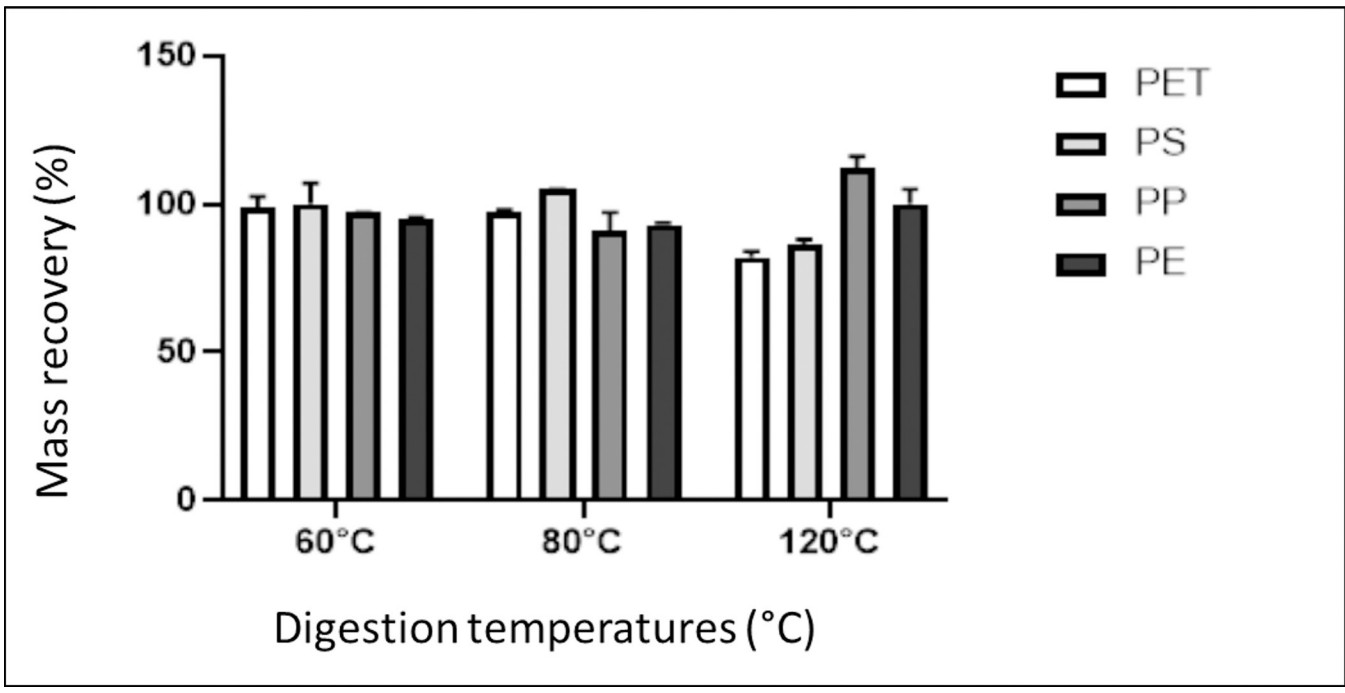

**Fig 5. Results of the mass percentage recovery of microplastics at different temperatures of Microwave-assisted digestion, using HNO₃/HCl mixture 3:1 v/v, for a time of 1 hour.**

digestion at 60°C is 100%, therefore, no experiments were conducted at higher temperatures. Raw data of Fig 7 can be found in S2 Table.

While it is true that cellulose mineralization was not quantitative for temperatures below 120°C, post-digestion chemical modification of the remaining cellulose did not generate a fluorescence when stained with NR, as shown in Fig 3.

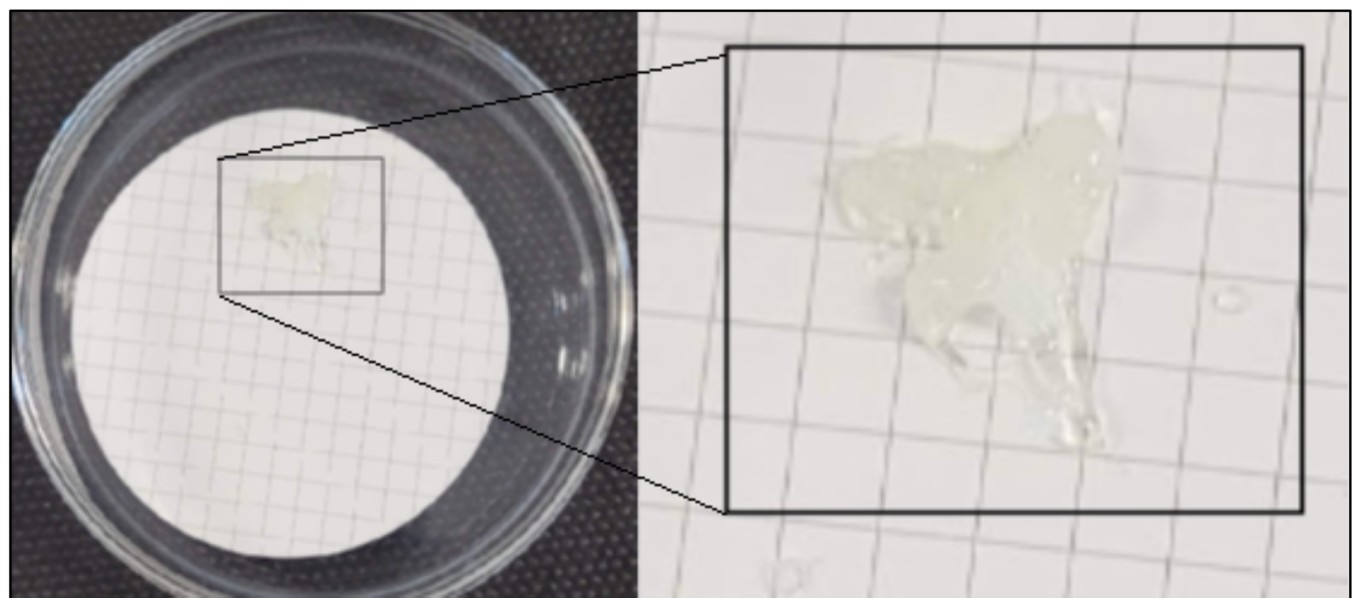

**Fig 6. Melted LDPE microparticles, because of microwave-assisted acid digestion, for one hour at 120°C.**

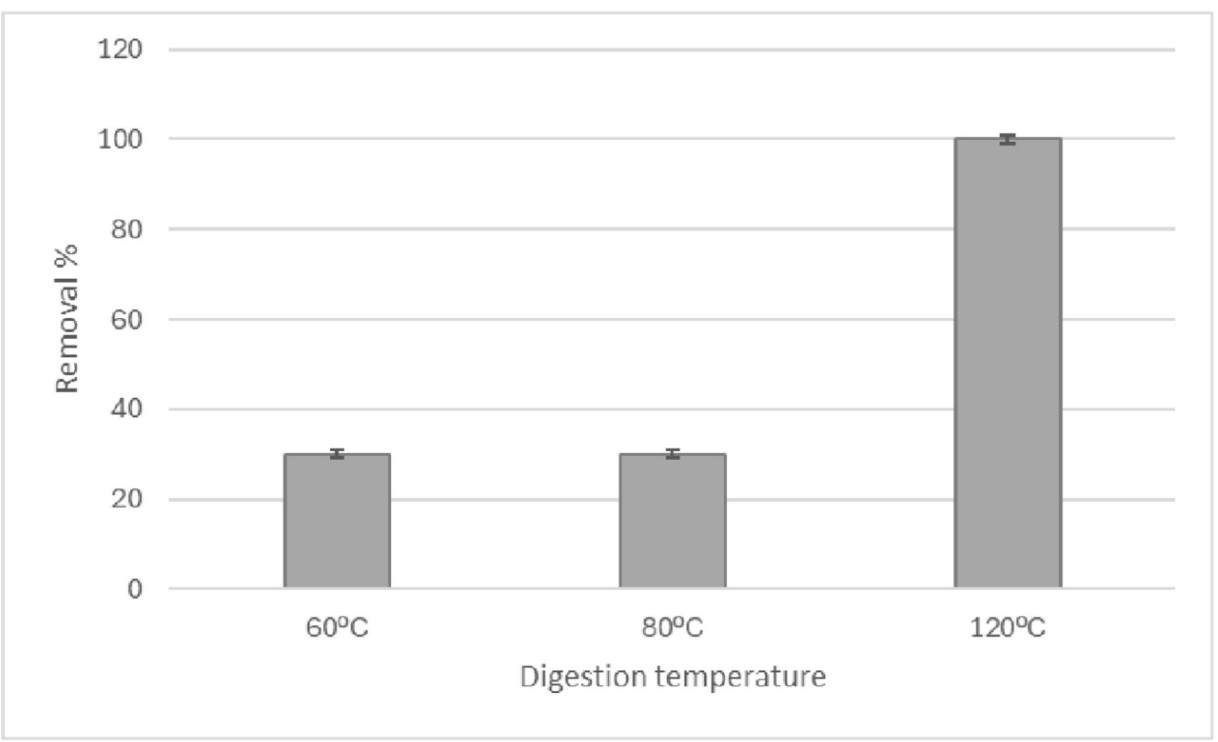

**Fig 7. Removal of cellulose, in the digestion stage, at three different temperatures.**

Considering the results of the recovery of MPs (Fig 5) and the removal chitin and cellulose (Fig 7) The best temperature that optimally balances the stability of PET, and the elimination of interferences is 60˚C. This decision agrees with the results obtained by Naidoo et al. [10], who conducted studies of the stability of MPs in nitric acid, to implement a digestion methodology in fish.

## Effect of the flotation step on accuracy for the determination of PET in microplastics

To demonstrate the necessity of incorporating a flotation step prior to detection, the most commonly used flotation methods from the literature were replicated. These methods have generally yielded reliable results for some of the microplastics mentioned earlier [3, 5, 6].

Two solutions widely used in the literature, NaCl (1.20 g cm$^{-3}$) and CaCl$_2$ (1.45 g cm$^{-3}$) are used in this experiment [22] which were carried out in triplicate, with the addition of 100 particles of each polymer to 5 g of purified sand [14]. The mixtures were combined with 30 mL of saturated saline and centrifuged at 3.900 rpm for 5 minutes. The supernatant is collected in a nitrocellulose membrane filtration system with a pore size of 0.45 μm followed by the application of the digestion method implemented in this study. Table 1 shows the results of flotation

**Table 1. Results of adding a pre-flotation step to the method using two saline's solutions.**

| Solution | % recovery of MPs | | | |
|---|---|---|---|---|
|  | **PET** | **LDPE** | **PP** | **PS** |
| NaCl (1,20 g cm$^{-3}$) | 0 | 92±2 | 96±3 | 97±0,6 |
| CaCl$_2$ (1,45 g cm$^{-3}$) | 98±1 | 90±9 | 98±1 | 95±2 |

with two types of solutions prior to digestion. Raw data of Table 1 can be found in S3 Table and image analysis for quantification in S1 Fig.

Information recovery essays during the flotation step are scarce. For example, Phuong et al. [3], analyse 70 the determination of MPs in marine sediments, 65 studies use the density extraction step, of which 8 studies report recoveries above 90% for those microparticles of lower density such as PP, PS and PE, but low recoveries PET (<90%) which is consistent with our results.

As expected, the method that includes a flotation step prior to digestion generally yields good recovery results. We observed that the best recoveries were obtained using the $CaCl_2$ solution, which, due to its higher density, achieved nearly quantitative recovery for the four types of plastics. In contrast, the NaCl solution, being less dense, resulted in poor recoveries for PET. It is important to clarify that the aim of reproducing the flotation step prior to digestion in this study is to assess the impact of omitting this step and performing direct sample digestion. Given the satisfactory results when using flotation with a $CaCl_2$ solution, it is anticipated that omitting this step and proceeding with direct microwave digestion of the wet sample may negatively affect the quality of the results, leading to lower recoveries if flotation is indeed crucial for separating plastic microparticles from interferences. On the other hand, if microwave digestion with acid is sufficient, recovery results should remain quantitative. Recovery studies for the direct digestion method will follow.

### Validation of direct digestion method

The Fig 8 shows the results obtained from the recovery study by counting PET microparticles through the selected digestion process, in three cases. Raw data of Fig 8 can be found in S4 Table and images used for MPs quantification in S2 Fig.

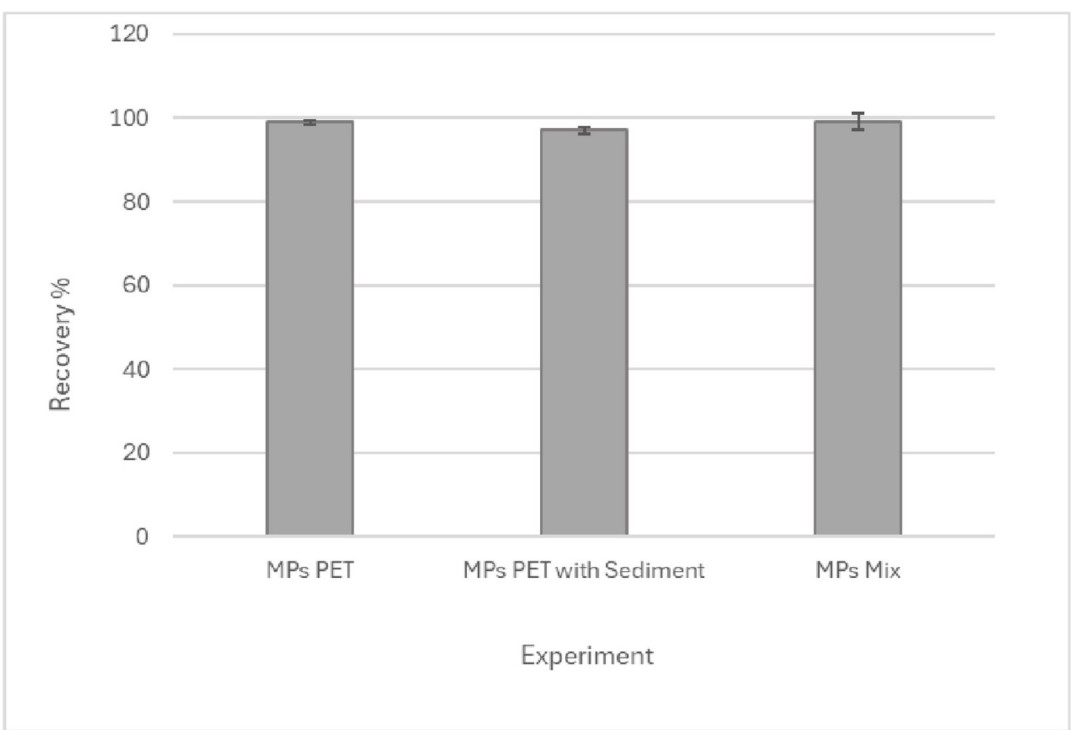

**Fig 8. Spike and recovery of PET microplastics (MPs) was conducted under the following digestion conditions: Isolated PET microplastic particles (MPs PET), PET microplastics spiked into marine sediments (MPs PET with sediments), and a mixture of microplastic polymers (MPs Mix).**

In the first case, recovery experiments were performed considering digestion only with PET MPs. As a second case recovery of PET MPs was performed spiking on 1 g real sediment sample. Finally, as a third case PET recovery were performed on a mixture of MPs of the four polymers used in this study was used.

When only the PET particles were subjected to digestion, a recovery of 99% was obtained, indicating that this type of microparticles have a high structural stability at the defined digestion conditions, ruling out possible fractionations. Similarly, when evaluating the effect of the matrix, when PET MPs were digested in the presence of sediment, a recovery of 97% was obtained. This demonstrates that the digestion and detection method together have sufficient selectivity to ensure that the constituents of the sediment matrix do not interfere with PET MPs counting. Finally, when evaluating the selectivity of the method when subjecting a mixture of MPs of different polymers to digestion, a high selectivity is observed when identifying PET MPs, with a recovery of 99% in the mixture of different polymers.

In accordance with the above, by evaluating the stability, matrix effect and selectivity, it can be confirmed that the proposed conditions of the method are adequate for the analysis of PET MPs in sediment when applying the digestion step directly on the matrix. No information has been found on literature about the use of microwave-assisted acid digestion in the analysis of MPs in sediment [3, 6]. In addition, the use of acid digestion has been described in some studies in seafood, but with a methodology in a different configuration than in our case and sometimes with poor results for the extraction of MPs from PET [7, 12]. In their study, Karami et al. [11] performed acid digestion tests at 25°C for 96 hours, with concentrated $HNO_3$ and HCl acids (used separately), obtaining PET recoveries of of 93.3% and 89.6% with $HNO_3$ and HCl respectively. To study the effect of digestion on the degradation of PET MPs, a recovery study was carried out in different masses. The results are presented in Fig 9, with recovery rates greater than 98% at each level studied, demonstrating an adequate resistance to degradation of

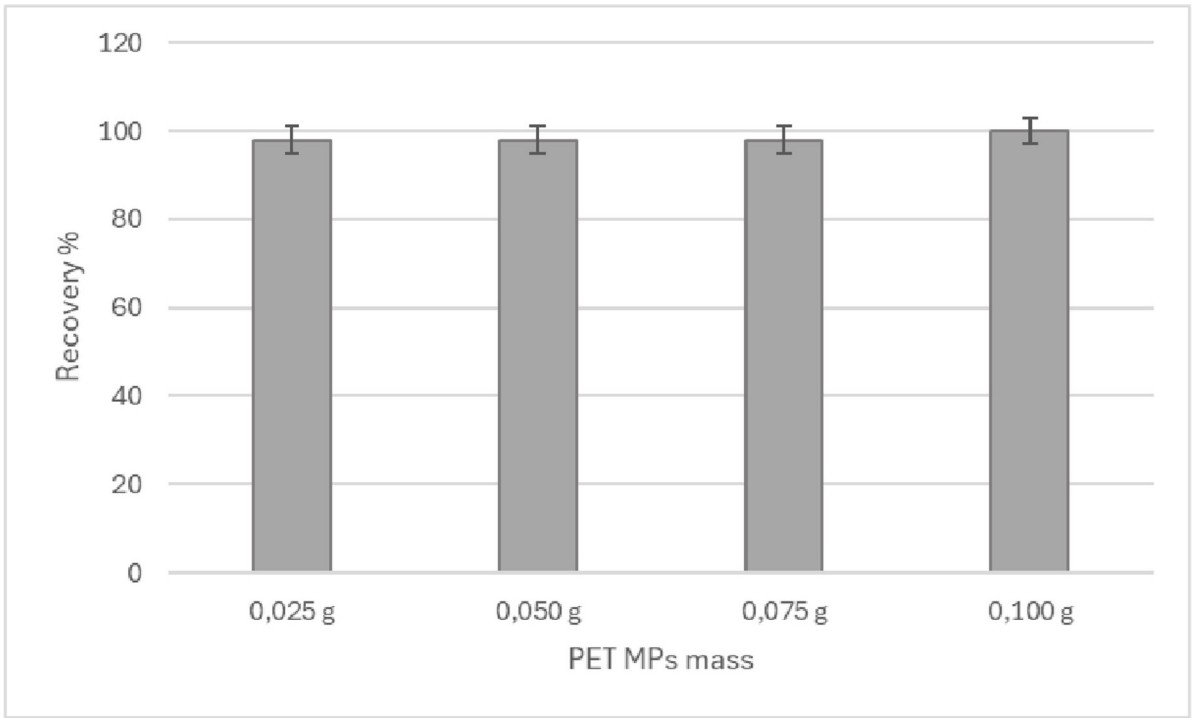

**Fig 9. Spike and recovery results at different levels of MPs PET masses.**

PET microparticles under the proposed digestion conditions, regardless of the amount present in the matrix, in the mass range studied. Raw data of Fig 9 can be found in S5 Table.

## Application of the methodology to PET monitoring in Loa River sediments

The Fig 10 shows the example of an image of Loa River´s sediment, where PET particles were identified by using Artificial Intelligence ChatGPT 4.0. The mass of one gram of sediment was treated with the methodology implemented in this study.

Analysis of Fig 10 shows that, by using artificial intelligence, it is possible to individually recognize the color ranges produced by microparticles from PET, PE, and PP standards. Additionally, we can observe that these ranges allow for the identification and marking of most particles from each standard in white. To evaluate the ability to selectively identify MPs in a real sample, the three RGB color ranges for each polymer were applied to a sample from the Loa River. The results are presented in Fig 11.

In Fig 11 (left), we can see that visually, pink range particles associated with the color of PET are detected. When applying the RGB color range, the artificial intelligence is able to recognize several particles marked in white, which correspond to those detected by the human eye. Subsequently, when the RGB color ranges for PE and PP are applied, very few particles are detected. As expected, several of these particles—indicated by arrows—are marked for both PE and PP, demonstrating that selective identification for these polymers still requires improvement before this method can be reliably extended to them. It is important to highlight that, although the recognition of PET particles is possible using artificial intelligence, to date, we

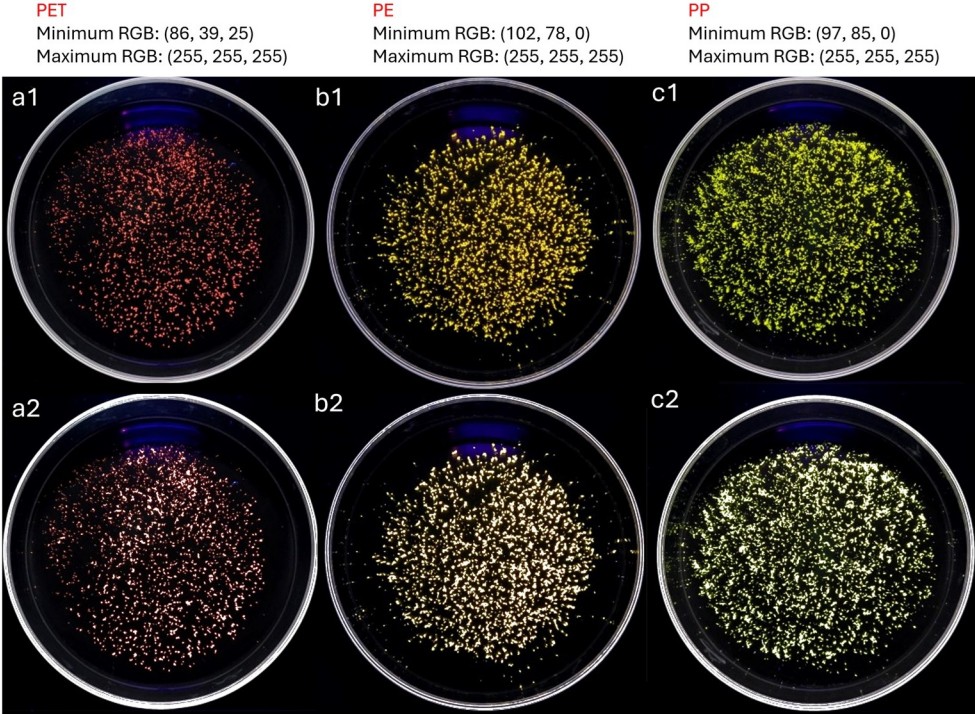

**Fig 10.** Identification of fluorescent emission colors for post-digestion microplastics standards of PET (a), PE (b), and PP (c) by using ChatGPT-4.0 artificial intelligence. The upper section provides the minimum and maximum RGB color range from IA color recognition for each plastic. To test the color range, images of fluorescent particles from each standard (1) were analyzed separately as samples using the same color range and each identified particle was marked in white (2).

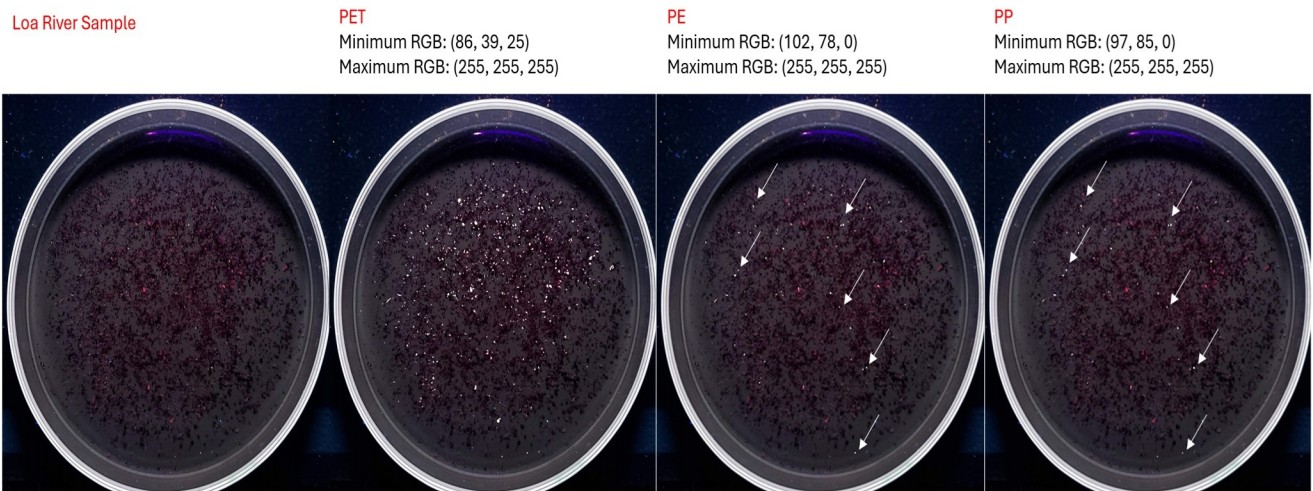

**Fig 11. Detection of PET, PE, and PP microplastic particles in a sediment sample from the Loa River using the RGB ranges proposed by the ChatGPT-4.0 artificial intelligence.**

have not yet been able to achieve automatic particle counting. This is due to the challenge of delineating the area of each particle so that it can be individually counted, an issue we have already reported in a previous study [23]. This limitation represents a task to be addressed in future research with more advanced programming solutions. As a result, fluorescent particle counting in this study continues to be performed manually.

It can observe in Fig 12 that the PET concentration levels in the samples from Taira are, on average, higher than those from Chiuchiu, Quillagua, and the River's mouth. When analyzing the sampling site map in Fig 1, this indicates that pollution is greater upstream. These results could be explained by the presence of floating plastic debris, as reported in a previous study by Honorato-Zimmer and collaborators, which was conducted in several rivers in northern Chile, including the Loa River. This study described that plastic waste accounts for 70% of the floating debris in these rivers, with an average of 62 items observed per hour. The primary sources of this pollution were identified as riverside users, local residents, and individuals illegally dumping litter [17]. Raw data for Fig 12 can be found in S6 Table, and the images used for PET microplastic quantification for each sample location along the Loa River are available in S3 Fig.

## Conclusions

From the results obtained in this study, we have demonstrated that by using microwave-assisted digestion applied directly to the wet sediment, steps used in sediment treatment, such as drying and sieving, are not necessary. Additionally, it was shown that density separation is not only unnecessary, but when applied it can lead to loss of PET particles. Therefore, it is concluded that this step should be eliminated from any study opting for direct acid digestion.

Microwave-assisted acid digestion at 60°C, applied directly to the wet sediment, is suitable in removing common matrix interferences such as cellulose and chitin, while ensures the stability of PET MPs.

The staining step, using NR as a fluorophore, after the implemented digestion method, is suitable for achieving satisfactory selectivity in the identification of PET MPs in the presence of interfering microparticles such as PP, PS and LDPE, under the conditions of the developed

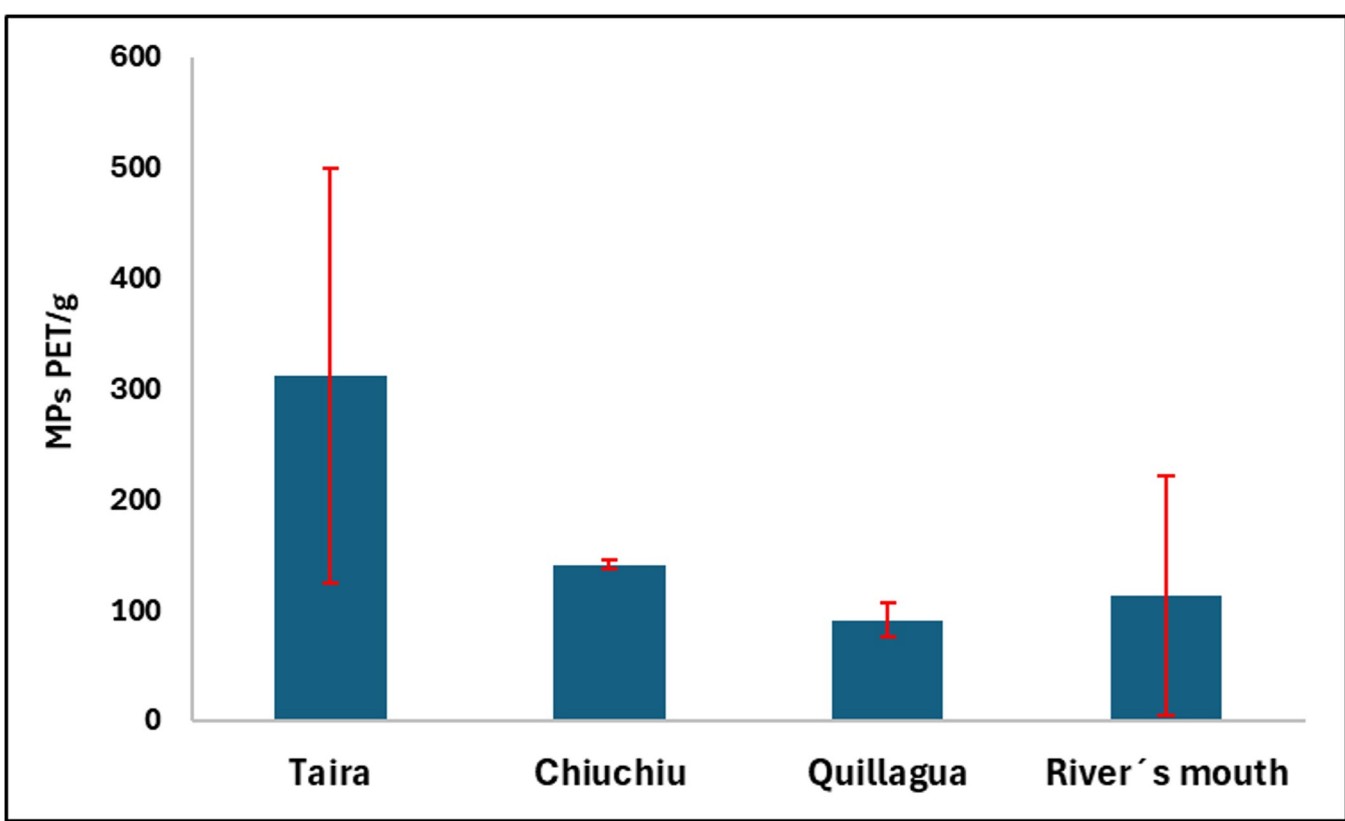

**Fig 12. Determination of PET microplastic particles in sediment samples from the Loa River in Chile (n = 3).**

method. Additionally, it allows for the counting of PET particles larger than 0.066 mm equivalent to 2 pixels.

Finally, this method proved useful for monitoring plastic microparticles in the sediments of the Loa River in Chile, revealing that the river is primarily contaminated with PET microparticles upstream in the Taira area, likely linked to plastic bottle pollution from tourists.

Looking ahead, as an outlook, the use of artificial intelligence for color recognition to identify different types of plastic polymers, as well as for the automatic counting of detected particles, holds great potential for expanding the scope of methods like the one presented in this study. However, there are still challenges to be addressed, such as optimizing the RGB color ranges required for selective identification and solving the issue of accurately delineating the boundaries of each particle to perform an accurate counting.

## Supporting information

**S1 File. Method for the determination of PET microplastics in sediment.**
(PDF)

**S1 Table. Fig 5. Values behind the means.** Results of the mass percentage recovery of microplastics at different temperatures of Microwave-assisted digestion, using HNO3/HCl mixture 3:1 v/v, for a time of 1 hour.
(XLSX)

**S2 Table. Fig 7. Values behind the means.** Removal of cellulose, in the digestion stage, at three different temperatures.
(XLSX)

**S3 Table. Table 1. Values behind the means.** Results of adding a pre-flotation step to the method using two saline's solutions.
(XLSX)

**S4 Table. Fig 8. Values behind the means.** Spike and recovery of PET microplastics (MPs) was conducted under the following digestion conditions: isolated PET microplastic particles (MPs PET), PET microplastics spiked into marine sediments (MPs PET with sediments), and a mixture of microplastic polymers (MPs Mix).
(XLSX)

**S5 Table. Fig 9. Values behind the means.** Spike and recovery results at different levels of MPs PET masses.
(XLSX)

**S6 Table. Fig 12. Values behind the means.** Determination of PET microplastic particles in sediment samples from the Loa River in Chile (n = 3).
(XLSX)

**S1 Fig. Image behind Table 1 values.**
(DOCX)

**S2 Fig. Fig 8. Image used for manual PET MPs quantification.** Spike and recovery of PET microplastics (MPs) was conducted under the following digestion conditions: isolated PET microplastic particles (MPs PET), PET microplastics spiked into marine sediments (MPs PET with sediments), and a mixture of microplastic polymers (MPs Mix).
(DOCX)

**S3 Fig. Image used for manual identification of PETs MPs in Fig 12.**
(DOCX)

## Author Contributions

**Conceptualization:** Marco Perez, Waldo Quiroz.

**Data curation:** Marco Perez, Cristofher Ferrada.

**Formal analysis:** Marco Perez, Sonnia Parra, Cristofher Ferrada, Waldo Quiroz.

**Funding acquisition:** Manuel Bravo, Waldo Quiroz.

**Investigation:** Marco Perez, Sonnia Parra, Pablo A. Perez.

**Methodology:** Marco Perez, Cristofher Ferrada, Pablo A. Perez.

**Project administration:** Waldo Quiroz.

**Resources:** Manuel Bravo.

**Supervision:** Sonnia Parra, Manuel Bravo, Waldo Quiroz.

**Validation:** Marco Perez.

**Writing – original draft:** Marco Perez.

**Writing – review & editing:** Waldo Quiroz.

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
