## [Decision Letter · Decision Letter 0]

6 Oct 2024

PONE-D-24-22018Development of a new methodology for the determination of PET microplastics in sediment, based on microwave-assisted acid digestion.PLOS ONE

Dear Dr. Quiroz,

Thank you for submitting your manuscript to PLOS ONE. After careful consideration, we feel that it has merit but does not fully meet PLOS ONE’s publication criteria as it currently stands. Therefore, we invite you to submit a revised version of the manuscript that addresses the points raised during the review process.

We look forward to receiving your revised manuscript.

Kind regards,

Amitava Mukherjee, ME, Ph.D.

Academic Editor

PLOS ONE

Journal Requirements:

“ANID-FONDECYT, Chile

Project 1230585.”

“The authors gratefully acknowledge financial support from ANID-FONDECYT, Chile

Project 1230585.”

“ANID-FONDECYT, Chile

Project 1230585.”

6. Please expand the acronym “ANID-FONDECYT” (as indicated in your financial disclosure) so that it states the name of your funders in full.

7. We note that your Data Availability Statement is currently as follows: [All relevant data are within the manuscript and its Supporting Information files.]

Reviewers' comments:

Reviewer's Responses to Questions

**Comments to the Author**

1. Does the manuscript report a protocol which is of utility to the research community and adds value to the published literature?

Reviewer #1: Yes

2. Has the protocol been described in sufficient detail?

To answer this question, please click the link to protocols.io in the Materials and Methods section of the manuscript (if a link has been provided) or consult the step-by-step protocol in the Supporting Information files.

The step-by-step protocol should contain sufficient detail for another researcher to be able to reproduce all experiments and analyses.

Reviewer #1: Partly

3. Does the protocol describe a validated method?

Reviewer #1: Yes

4. If the manuscript contains new data, have the authors made this data fully available?

Reviewer #1: Yes

**5. Is the article presented in an intelligible fashion and written in standard English?**

Reviewer #1: **No: **The overall flow of the manuscript must be improved.

6. Review Comments to the Author

Reviewer #1: The overall methodology focuses on the recovery of PET particles. Kindly check the comments which have been included as an attachment.

7. PLOS authors have the option to publish the peer review history of their article (what does this mean?). If published, this will include your full peer review and any attached files.

Reviewer #1: No

---

## [Author Response · Author response to Decision Letter 0]

24 Oct 2024

Reviewers' comments:

Reviewer's Responses to Questions

Comments to the Author

5. Is the article presented in an intelligible fashion and written in standard English?

Reviewer #1: No: The overall flow of the manuscript must be improved.

R: Several grammatical errors were corrected, and the overall flow of the text was improved by incorporating connecting paragraphs and repositioning entire sections. We believe this new structure better follows the logical sequence of the validation process: first reproduction of staining Nile Red method, second the sample treatment validation; third, the color identification process; and finally, the application of the method to real samples.

6. Review Comments to the Author

The authors have formulated a way to recover PET particles from the sample by subjecting the

samples to differential conditions, including different temperature and excitation wavelength.

These conditions have successfully eliminated other interferences at each consequent stages

and resulted in the detection of PET MPs. I would like to suggest the following:

General comments: Applies to the entire manuscript.

1. Check the manuscript for grammatical and spelling errors. For example: In keywords

“Red Nyle” is not correct.

R: Red Nyle was corrected to Nye Red or NR in all manuscript

2. For all the statistical representation, mention the software used for the preparation of

graphs. Additionally, include a statistical analysis section for this in the methodology

head.

R: The "Statistical Analysis" section included details on the software used, the type of statistical tests, and the confidence level applied for each data analysis.

3. I suggest it would be appropriate if a pictorial representation or a flowchart is included

for methodology, including the effect of each step (such as elimination of chitin

fluorescence after digestion)

R: Fig. 2. Flowchart of the method was included according to reviewer advice. Representative pictures and the explicit purpose of each analytical step were included. This figure replace Figure 1 on this new version of the manuscript

4. The font consistency is lacking. For example: “Result and Discussions” and

“Conclusions” are in different forms; also check font in Table 1.

R: Font and size font was checked on all manuscript including head titles, table and figure labels “Vancouver font” was used in all manuscript and the size of the font and formatting were corrected according to Plos One “Manuscript body formatting guidelines”,

Hence, I recommend the authors to rewrite the manuscript considering all the above stated

suggestions and thus improving the overall flow of the manuscript.

Specific comments:

Major comments-

1. The addition of the possible real-life applications of this methodology would be

beneficial for the readers. Additionally, the methodology referred to in reference

number 9 has successfully recovered five polymer types. I suggest the authors to clearly

include a reason for their focus only on PET.

R: At the end of the discussion, an explanatory paragraph was added to clarify why this study focused on PET. Briefly, there are two main reasons. First, PET is one of the most frequently detected plastics in municipal waste. Locally, a published study has shown that many floating plastic containers have been found in Chilean rivers, including the Loa River, which is part of our study. Finally, our results indicate that this type of polymer was the most frequently observed in these samples. 3 references were included to support our choice. Finaly a monitoring was performed along sediments of Loa River in Chile. 4 sampling locations were included in this new version and results are presented in Figure 12.

2. It would be effective if the authors stated the reasons for each and every step as it will

provide great understanding for the readers.

R: This suggestion aligns with that made by the other reviewer. We recognize the importance of clarifying the purpose of each step in the methodology. For this reason, we have removed Figure 1 and replaced it with a pictorial flow diagram that includes the description and purpose of each methodological step. This is now Figure 2 in the revised manuscript.

3. Line 130- It would be good if the authors explained the use of only 254nm wavelength

filter and not the 365nm wavelength as mentioned in the reference paper. 

R: The results of the Nile red-stained microplastics exposed to a 365 nm excitation wavelength were included, along with a discussion demonstrating that selective fluorescence was not achieved at this wavelength. This is now Figure 4 in the revised version of the manuscript.

Also state the efficiency of this method over already existing methods. Additionally, clarification of the possibility to recover other particles using different excitation wavelengths would be an added advantage. This information would be helpful since many researchers aim

to isolate all the potential plastic constituents from the environmental samples. Again,

I suggest the authors to clearly include a reason for their focus only on PET. For this

addition of microscopic images depicting the particles in bright field, the selected

fluorescence wavelength (254nm) and in other wavelengths, superimposed on each

other would substantiate the present study.

R: We greatly appreciate this reviewer’s comment, as it highlights the broader potential of the method we presented in the initial version of the manuscript. Indeed, this method has the potential to be extended to the identification and quantification of PE and PP. The reason we focused solely on PET is because this method was designed to be applied to the Loa River, which is primarily contaminated by PET plastic bottles, as described in a previous study that has been added to the introduction.

However, the identification of PE and PP still faces a selectivity challenge that we have not yet fully resolved. As shown in Figure 3, both PE and PP fluorescence colors are yellow with similar tones. Nevertheless, we agree with the reviewer that it is important to make readers aware of the real potential of this method.

Therefore, we have taken two actions: First, we used RGB color recognition through the artificial intelligence platform ChatGPT-4.0 to demonstrate two key points. Firstly, the method selectively detects PET. Secondly, while AI-based color recognition is possible for PE and PP, selectivity remains a challenge that must be addressed to fully unlock the method's potential.

For this reason, we have included Figure 9 to demonstrate the power of RGB color recognition for the three plastics (PET, PE, and PP) and Figure 10 to show the method’s selectivity for detecting PET, as well as the limitations regarding selectivity for PE and PP

Minor comments-

1. Considering that the density of PET is less than that of CaCl2, causing it to float, and

that LDPE, PP, and PS are recovered in both solutions due to their lower density, I have

a question regarding the use of NaCl (1.20 g/cm3). Since PET would settle in NaCl at

the above density, I would like to suggest the authors to consider recovering the PET

in a denser NaCl solution. If not, it would be good if explanation was provided for why

this approach was not considered for the recovery of PET in NaCl.

R: We acknowledge that presenting the flotation step results after the direct digestion method may give the impression that this step was not sufficiently optimized. However, the purpose of reproducing the flotation step is not to incorporate it into our procedure, but rather to demonstrate that it is unnecessary and that good recovery results can still be achieved by omitting this step and directly applying microwave digestion to the sample. To avoid further confusion, the flotation method results were presented first and discussed in this context, prior to introducing the recovery results from the direct digestion method.

2. Mention the purpose of the rapid cooling of the samples in line 126 and also provide

the temperature.

R: By apply this cycle of 50°C for staining and then 0°C for cooling have shown us that the particles exhibit more intense and longer-lasting fluorescence. We include the temperature and a proper discussion about this process.

3. Line 146: Briefly describe the calcination process and why it is used here?

R: Considering the heterogeneity in the distribution of microplastics in the environment, spike and recovery studies on real samples cannot be conducted using naturally contaminated samples because “native microplastics” cannot be subtracted mathematically. For this reason a calcination procedure was applied to a real sediment sample, to eliminate native MPs. A proper explanation was included at the end of “Trueness determination” method description according to reviewer advice.

---

## [Decision Letter · Decision Letter 1]

12 Nov 2024

Development of a new methodology for the determination of PET microplastics in sediment, based on microwave-assisted acid digestion.

PONE-D-24-22018R1

Dear Dr. Quiroz,

We’re pleased to inform you that your manuscript has been judged scientifically suitable for publication and will be formally accepted for publication once it meets all outstanding technical requirements.

Kind regards,

Amitava Mukherjee, ME, Ph.D.

Academic Editor

PLOS ONE

Additional Editor Comments (optional):

Reviewers' comments:

Reviewer's Responses to Questions

**Comments to the Author**

1. Does the manuscript report a protocol which is of utility to the research community and adds value to the published literature?

Reviewer #1: Yes

2. Has the protocol been described in sufficient detail?

To answer this question, please click the link to protocols.io in the Materials and Methods section of the manuscript (if a link has been provided) or consult the step-by-step protocol in the Supporting Information files.

The step-by-step protocol should contain sufficient detail for another researcher to be able to reproduce all experiments and analyses.

Reviewer #1: Yes

3. Does the protocol describe a validated method?

Reviewer #1: Yes

4. If the manuscript contains new data, have the authors made this data fully available?

Reviewer #1: Yes

**5. Is the article presented in an intelligible fashion and written in standard English?**

Reviewer #1: Yes

6. Review Comments to the Author

Reviewer #1: Most of the explanations are appropriate and the article has been refined. This article is noow helpful to thee scientific community.

7. PLOS authors have the option to publish the peer review history of their article (what does this mean?). If published, this will include your full peer review and any attached files.

Reviewer #1: No

---

## [Editor Report · Acceptance letter]

27 Nov 2024

PONE-D-24-22018R1 

PLOS ONE

Dear Dr. Quiroz, 

I'm pleased to inform you that your manuscript has been deemed suitable for publication in PLOS ONE. Congratulations! Your manuscript is now being handed over to our production team.

Kind regards, 

on behalf of

Professor Dr. Amitava Mukherjee 

Academic Editor

PLOS ONE